# High-Impact Risk Factors for *Mycobacterium avium* ssp. *paratuberculosis* in Dairy Herds in Germany

**DOI:** 10.3390/ani13121889

**Published:** 2023-06-06

**Authors:** Mette Krieger, Susanne Eisenberg, Karsten Donat, Amely Campe

**Affiliations:** 1Department of Biometry, Epidemiology and Information Processing, WHO Collaborating Centre for Research and Training for Health at the Human-Animal-Environment Interface, University of Veterinary Medicine Hannover, 30559 Hanover, Germany; amely.campe@tiho-hannover.de; 2Lower Saxonian Animal Diseases Fund, 30169 Hanover, Germany; susanne.eisenberg@ndstsk.de; 3Animal Health Service, Thuringian Animal Diseases Fund, 07745 Jena, Germany; kdonat@thtsk.de

**Keywords:** *Mycobacterium avium* ssp. *paratuberculosis*, risk factor, dairy herd, odds ratio, Johne’s disease

## Abstract

**Simple Summary:**

Previous studies have investigated the role of different factors in paratuberculosis introduction in dairy farms or dairy cattle. Because paratuberculosis between-herd prevalence was shown to differ substantially between three regions of Germany, this study evaluated different management factors for their impact on the paratuberculosis status of a farm. The most obvious impact was found for an increase of 100 or more cows per herd, followed by the purchase of cattle with unknown paratuberculosis status and limitations in calf feeding management within the barn. These aspects should be prioritized in paratuberculosis control beneath a tailored control approach for individual farms and regions.

**Abstract:**

In a cross-sectional study, it was identified that three regions in Germany differed with respect to their herd-level prevalence for paratuberculosis in dairy cattle. In the study presented here, the same farms were analyzed to identify those components of biosecurity and farm management with the highest impact on *Mycobacterium avium* ssp. *paratuberculosis* (MAP) introduction and establishment in a farm. Hence, the data analyzes included 183, 170 and 104 herds from the study regions north, east and south, respectively. A herd was considered MAP-positive if at least one fecal environmental sample was positive. Twenty-six different possible risk factors from five different components of biosecurity and farm management were analyzed. We show that the average management of calf feeding increased the odds for a MAP-positive farm by 5.22 times (95% confidence interval (CI) = 1.25–21.83). With every 100-cow increase in farm size, the risk for a farm to test MAP-positive increased by 1.94 times (CI = 1.15–3.27), 1.14 times (CI = 1.02–1.27) and 5.53 times (CI = 0.44–68.97) in the north, east and south study regions, respectively. Furthermore, the purchase of cattle with an unknown MAP status increased the risk for a farm testing MAP-positive by 2.86-fold (CI = 1.45–5.67). Our results demonstrate that herd size, unknown MAP status of the purchased cattle and different aspects of calf feeding play an important role in the MAP status of a farm and should be in focus in regions with different MAP between-herd prevalence. Additionally, farm individual risk patterns should be identified during (veterinary) biosecurity consultancy.

## 1. Introduction

In previous studies, several biosecurity and farm management factors were identified as risk factors for paratuberculosis infection in a herd. Herd size [1,2,3] and animal purchase [3,4,5] play an important role in the risk of a herd testing *Mycobacterium avium* ssp. *paratuberculosis* (MAP)-positive. Also other risk factors regarding outdoor hygiene were identified as risk factors for paratuberculosis, for example, keeping heifers on contaminated pastures increased MAP excretion within infected herds [4], and keeping animals other than cattle on the farm could be a source of infection [6,7]. Besides many management factors which are recommended for paratuberculosis control, hygiene of equipment and animals is also considered a useful tool on a farm [8]. Furthermore, several studies have investigated the role of calf management for MAP transmission within a herd. The contact of calves with the feces of adults was shown to be the most important factor for transmission within a herd [9], and suckling of the dam’s udder by calves increased the risk for a positive MAP status of the herd [3,10]. Furthermore, it was shown that the use of calving pens to house sick or lame cows [11] and group housing of preweaned calves [12,13] increased the risk of MAP transmission and infection. The evaluation of calf feeding revealed that MAP can also be shed into colostrum and milk [14,15], and fecal contamination of milk is possible [16].

Herd-level diagnosis of the MAP status by fecal culture or polymerase chain reaction (PCR) of fecal environmental samples has been described as an appropriate tool [17,18]. In addition, combining PCR and fecal culture of boot swabs and liquid manure samples [19] or of environmental fecal samples from different locations [20] results in an accurate detection of paratuberculosis-positive herds.

The MAP status in herds enrolled in this study was determined using a standardized sampling scheme for environmental samples, liquid manure and sampling socks [21]. Different values of between-herd prevalence for three different regions of Germany (apparent prevalence: study region north = 12.0%, east = 40.6% and south = 2.9%; corrected true prevalence: study region north = 14.8%, east = 50.1%, south = 3.6%) were identified. Each region reflects different agricultural structures, with a high animal density in region north, large dairy herds with a low farm density in region east and a high farm density with small herds in region south [22].

The objective of this study was to identify those components of biosecurity and farm management that have the highest impact on MAP introduction and establishment in a farm. Furthermore, noticeable risk factors are to be described in order to derive a tailored control approach for the investigated farms and regions.

## 2. Materials and Methods

### 2.1. Study Design

The herds were located in Germany, and the analysis was conducted according to a random stratified sampling scheme for each region. The data originated from a cross-sectional study conducted between 2016 and 2020 [21,23]. In the PraeRi study (Animal health, hygiene and biosecurity in German dairy cow operations—a prevalence study [23]), the sample size was set at 250 farms per region (power 80%, significance level 5%), calculated using NCSS PASS Version 13.0.8 [24]. The dairy farms in the study region north and east were randomly selected from the German database of animal identification and registration, stratified by region and herd size [23]. In the study region south, the dairy farms were randomly selected from dairy farms organized in the dairy herd improvement organization for Bavarian dairy farms (Milchprüfring Bayern e.V., Wolnzach, Germany), stratified by region and herd size as well [23]. Our study was part of the PraeRi study, and only those farms that agreed to be sampled for paratuberculosis were analyzed. The farms were visited by study veterinarians to evaluate farm characteristics such as animal health, biosafety, feeding and animal housing on a farm.

### 2.2. Analyzed Data

Twenty-six variables were analyzed as possible risk factors for paratuberculosis infection at the herd level. Some of these variables were measured at the animal level (e.g., breed and hygienic score of cows and calves), and other variables (e.g., regarding feeding, animal housing and biosafety) were evaluated with an interview and data entry forms at the herd level or at a stable compartment level [23]. Variables that were measured at the animal level were aggregated on herd-level values, i.e., percentage of breed and hygienic scores for a farm.

As we aimed to demonstrate from which component of biosecurity and farm management (BFM) the highest risk for MAP infection and transmission may originate, we assigned the variables of interest to different components of BFM.

General farm management practices were sorted as internal biosecurity and external biosecurity, and calf management practices were sorted as calving, calf housing and calf feeding with the assigned variables of interest (Figure 1).

The variables were scored as “good” (reference category), “average” and “insufficient” regarding the level of hygienic quality (i.e., MAP prevention potential). The quantitative variables, except herd size, were scored likewise using their 33% and 66% quantiles as cut-offs for each region. The modus of the quality scores of all respective variables was then used to assign an overall hygienic quality score to each component of BFM (i.e., “good”, “average” and “insufficient”).

The housing type (≥80% of cows were housed in the according housing type) and herd size (i.e., current number of dry and lactating cows) were analyzed as possible confounders in multivariable regression models. Herd size was included as a quantitative variable.

Paratuberculosis samples were collected between July 2017 and July 2019. Detailed information regarding environmental sampling and laboratory analysis has been published previously [21]. The MAP herd status was determined by PCR and fecal culture. Farms with at least one positive environmental sample, determined by PCR and/or fecal culture, were classified as MAP-positive.

### 2.3. Statistical Analysis

The data were analyzed with SAS^®^ software, Version 9.4 (SAS Institute Inc., Cary, NC, USA). The statistical unit was the herd. A descriptive analysis was performed stratified by region and MAP status. To evaluate the association and the strength of the association between categorical variables, Fisher’s exact test (cut off: ≤0.05) and Cramer’s V test (cut off: >0.5 for strong association) were used. To evaluate the association between categorical and quantitative variables, Kruskal–Wallis test (cut off: ≤0.05) and Wilcoxon two-sample test (cut off: ≤0.05) were used. Spearman’s rank correlation coefficient (cut off: >0.7) was used to evaluate the strength of the association between quantitative variables. If significant associations were identified and the descriptive analysis underlined the association between variables, only one variable was included in the multifactorial model. We used univariable and multivariable logistic regression models to assess the association between risk factors (independent variable/s) and MAP status (dependent variable). For each of the three regions, one separate model was fit. Univariable logistic regression analysis was performed for each variable. In study region south, herd prevalence and sample size were too small for model fitting including categorical variables. Therefore, we performed only descriptive analyses for categorical variables in study region south. Farms with missing data for an independent variable were not included in the univariable analyses.

Multivariable logistic regression was only performed for components of BFM. One multivariable analysis was performed with general farm management and calf management. The other multivariable analysis was performed with internal biosecurity, external biosecurity, calving, calf housing and calf feeding as a stepwise backward selection. Here, external biosecurity and calving were not allowed for selection due to their contextual importance for the entrance and establishment of MAP in a farm. Although housing type and herd size were to be analyzed as possible confounders, only herd size was finally included in the models. This was due to the high association between herd size and housing type. Herd size was included as a confounder in all models, considering parameter estimates and model fit criteria to identify the best fitted model. The models with herd size showed no effect on the parameter estimates of the other included variables, and lower values of the model fit criteria were reached.

To estimate the proportion of MAP-positive farms in exposed farms and in the population, the attributable fraction exposed (AFe) and the population attributable fraction (AFp) were also calculated.

## 3. Results

A total of 660 dairy farms (study region north: 199 farms, study region east: 201 farms, study region south: 260 farms) were invited to participate in the study, out of which 457 farms (69.24%) participated and could be analyzed, with 183 farms (91.96%) in the study region north, 170 (84.58%) in the study region east and 104 (40%) in the study region south.

### 3.1. Herd Size and Housing Type

The mean herd sizes in tie stalls were 33 cows (study region north; *n* = 7 farms), 28 cows (study region east; *n* = 3 farms) and 22 cows (study region south; *n* = 33 farms). The mean herd sizes in free stalls were 104 cows (study region north; *n* = 154 farms), 331 cows (study region east; *n* = 129 farms) and 54 cows (study region south; *n* = 68 farms). The herd size was associated with the housing type (*p* < 0.01 in Wilcoxon two-sample test for the study regions north, east and south). With every 100-cow increase in farm size, the risk for a farm to test MAP-positive increased by 1.94 times (95% confidence interval (CI) = 1.15–3.27, *p* = 0.01), 1.14 times (CI = 1.02–1.27, *p* = 0.02) and 5.53 times (CI = 0.44–68.97, *p* = 0.18) in the north, east and south study regions, respectively (Figure 2; Table 1).

### 3.2. General Farm Management

#### 3.2.1. Internal Biosecurity

In multivariable analysis of biosecurity components, farms with insufficient internal biosecurity in the study region east had a 1.72-fold higher chance of becoming MAP-positive compared to farms with good internal biosecurity (*p* = 0.05; Table 2). We found that 52% of MAP cases in the eastern population were due to bad internal biosecurity and could have been prevented, if internal biosecurity had been better (Table 3).

#### 3.2.2. External Biosecurity

In univariable analysis, animal purchase without inquiring about the MAP status increased the chance of a farm testing MAP-positive by 2.86 times in the study region east (CI = 1.45–5.67, *p* = 0.02) compared to no purchase, and 44% of the MAP cases in MAP-positive farms could have been prevented, if a farm had not purchased cattle (see Appendix A). This effect could not be detected in the other regions. In contrast, in the other regions, animal purchase with inquiring about the MAP status increased the chance for a MAP-positive farm compared to no purchase by 1.97 times (CI = 0.62–6.26; *p* = 0.15; region north) or by 1.28 times (CI = 0.43–3.81; *p* = 0.6; region east) (see Appendix A).

### 3.3. Calf Management

#### 3.3.1. Calving

If the calf stayed with the mother after birth, the time (h) of contact was evaluated. The suckling time of the calves differed between regions (33% quantile = 6 h in study region north, 8 h in study region east, 1 h in study region south; 66% quantile = 24 h in study region north, 12 h in study region east and south). The suckling of calves reduced the risk for being tested MAP-positive in the eastern region by 0.57 times (CI = 0.31–1.07; *p* = 0.08; see Appendix A) compared to farms where the calves were not allowed to stay with their dam.

#### 3.3.2. Calf Housing

Within univariable analysis of calf rearing, farms in the study region east that reared calves from other farms had a 7.86-fold higher chance (CI = 0.9–68.82; *p* = 0.16) for a MAP-positive status than farms that reared only their own calves. In addition, 53% of the MAP cases in MAP-positive farms were due to foreign calves and could have been prevented, if calf rearing had only been accomplished from the own herd (see Appendix A). Farms in the study regions north and east with no calf rearing (i.e., rearing of own or purchased calves until weaning or the age of six months) had a 3.69-fold higher chance (CI = 0.63–21.5; *p* = 0.09) in the study region north and a 1.57-fold higher chance (CI = 0.1–25.58; *p* = 0.7) in the study region east for a MAP-positive status compared to farms that only reared their own calves (see Appendix A). We found that 64% of the MAP cases in MAP-positive farms in the northern region could have been prevented, if calf rearing had have better organized (see Appendix A).

#### 3.3.3. Calf Feeding

In multivariable analysis, farms in the study region north with average calf feeding had a 5.22-fold higher chance (CI = 1.23–21.83) for a MAP-positive status (*p* = 0.02) than farms with a good calf feeding management. Farms in the study region north with insufficient calf feeding had a 1.6-fold higher chance (CI = 0.4–6.34) for an MAP-positive status compared to farms with good calf feeding (*p* = 0.6; Table 2). In univariable analysis, the results were similar. It appeared that 59% of the MAP cases in the northern population were due to average calf feeding and could have been prevented, if calf feeding had been better (Table 3).

The cleaning of buckets less often than after every feeding increased the risk for a farm to test MAP-positive. Buckets that were not cleaned at all increased the risk for a farm to test MAP-positive by 3.4 times (CI = 0.5–23.03; *p* = 0.63 in study region north) and by 1.86 times (CI = 0.62–5.62; *p* = 0.78 in study region east; see Appendix A). Buckets that were cleaned only once a day increased the risk for a MAP-positive farm by 4.58 times (CI = 1.09–19.17; *p* = 0.15 in study region north) and by 3.51 times (CI = 1.37–9.0; *p* = 0.06 in study region east). In addition, 74% of the MAP cases in MAP-positive farms could have been prevented, if the buckets had been cleaned more often than once a day (study region north; see Appendix A).

## 4. Discussion

This study aimed to evaluate the effect of different components of biosecurity and farm management on the MAP herd status in the three regions of Germany that are characterized by different MAP between-herd prevalence rates [21] and different agricultural structures. Furthermore, we wanted to illustrate high-impact risk factors for MAP. Due to different agricultural characteristics in Germany [22], the analysis was stratified into three regions. Hence, we wanted to investigate possible infection pathways in herds of these specific regions to offer tailored advice for disease control. Because of the different MAP between-herd prevalence [21] and agricultural structures [22], the results of this study can be transferred to other regions of the world with similar between-herd prevalence and agricultural structures.

Recommendations to control MAP in a herd generally consist of a combination of measures to reduce the risk of infection in a certain age group [8,25,26]. Therefore, it can be assumed that single risk factors play only a minor role in the development of paratuberculosis on a farm. The improvement of several management measures at the same time will potentiate the effect compared to one single improved step. Furthermore, from a population perspective, it might be difficult to identify a certain set of risk factors responsible for the most MAP-infected herds within a region, which is what we experienced during the study presented here. Therefore, we investigated different components of BFM and could identify calf feeding as an important farm management component which should be considered, especially, in MAP-positive farms.

### 4.1. Herd Size and Housing Type

We demonstrated that larger herds are more likely to test MAP-positive (Figure 2), which is in accordance with previous studies [1,2,3]. The farms in study regions north and south generally seemed to have a higher risk of testing positive compared to those in study region east, given a certain herd size. However, at the time of sampling, the herds in regions north and south were on average smaller than those in farms in study region east. If farms in study region north or south increase their herd size to similar values as in study region east, a drastic increase in the risk of becoming MAP-positive should be anticipated. The reasons for an increased risk of testing MAP-positive with an increase in the herd size, especially in study regions north and south, can only be hypothesized. One reason might be different hygiene or other farm management factors which are introduced when the herd size increases. It is possible that farms with larger herds purchase more cattle or that these herds include more cattle with unknown paratuberculosis status [3]. Furthermore, at the same time, large herds have both more animals shedding the organism and more animals at risk for infection than small herds, even if the within-herd prevalence is similar. Furthermore, the possibility for identifying a MAP-positive animal increases with an increase in the herd size if more animals are tested in larger herds [3]. Vilar et al. [1] discussed that contact between young and adult animals and, therefore, with MAP-contaminated feces occurs more frequently in larger herds because of missing paddocks to separate age groups from each other. In larger herds, the risk for transmission could increase because of multiple cows calving together, with imperfect calving hygiene and weak hygiene in calf rearing [27]. However, these factors were not investigated here. On the other hand, we have to consider that the smaller sample size and the low number of MAP-positive farms in study region south might have led to more imprecise estimates compared to study regions north and east [28]. However, regardless of precision, herd size seems to play an important role in predicting the MAP status of a herd in Germany as well as in other countries. Therefore, the ongoing agricultural changes characterized by a reduction in the number of farms, on the one hand, and an increase in herd size, on the other hand, should motivate the improvement of MAP control in the future.

As in previous studies [29,30], herd size was associated with housing type in our study. Tie stalls are less common than free stalls but are more commonly found in farms with smaller herds than in those with large herds. In general, most of the cows in the study region north are kept in free stalls (84.15%) compared to those (65.38%) in the study region south [21]. Moreover, herds in free stalls have a higher MAP between-herd prevalence than herds kept in tie stalls [29]. This may be due to reduced contact between cows in tie stalls compared to those in free stalls [29] or to different fecal contamination patterns in these different housing types [30]. However, a higher risk of testing MAP-positive for free-stall herds compared to tie-stall herds could not be detected in this study.

### 4.2. General Farm Management

#### 4.2.1. Internal Biosecurity

The hygiene of housing, equipment and cows is important for paratuberculosis control [8]. By including the hygiene of boots and hands, lying areas and walkways, as well as the hygiene of the udders in our analysis, hygienic aspects of equipment and animals were considered. None of these aspects of internal biosecurity appeared to have a considerable impact on the MAP status in the study presented here. This might be explained by the fact that these indisputably useful measures alone do not protect susceptible animals such as calves and might therefore be of less importance when compared with all components of BFM.

#### 4.2.2. External Biosecurity and Calf Housing

One of the general recommendations to prevent the introduction of paratuberculosis into a herd is to close the herd by rearing only its own calves and youngstock for replacement [31]. As this was practiced by most of the farms in all regions, our results support this recommendation. As already shown elsewhere [3,4], our results support the possibility that the purchase of (adult) cattle with an unknown MAP status poses a risk for MAP infection, explaining part of the higher MAP between-herd prevalence in the study region east. Furthermore, this is also important for the introduction of calves from other calves into a herd. This effect was most obvious in study region east, where the odds for a MAP-positive status were increased by 7.86 times (CI: 0.9–68.82; see Appendix A) when own calves and calves from other farms were reared in the respective farms. Therefore, farmers should inquire about the MAP status of purchased animals before introducing them into their herd, as recommended for paratuberculosis control [8,25,26]. In addition, also animal purchase after inquiring increased the risk of a herd to test MAP-positive; therefore, animal purchase should be avoided whenever possible.

### 4.3. Calf Management

#### 4.3.1. Calving

Surprisingly, our data suggest that suckling calves for a certain time after birth seems to decrease the MAP risk for a herd. It is known that calves can take up infectious fecal material from the dam’s udder [10]. Ansari-Lari, Haghkhah, Bahramy and Baheran [10] determined an odds ratio of 6.4 for a herd to test MAP-positive, if the udders of periparturient cows were contaminated with feces compared with no contamination. Besides the fecal contamination of colostrum, MAP has been shown to be shed directly into colostrum in clinical stages of the disease [14]; therefore, colostrum might play a role in MAP transmission between dams and their offspring [32]. It is common agricultural practice in some German herds to have the calf stay with the cow until the next milking and feeding time for reasons such as efficiency and personnel resources. Hence, any changes from this might indicate reverse causation in the way that calves are kept from a contact with the cow for hygienic reasons in a MAP control attempt. We assume that the farmers were already aware of the disease and had an ongoing or initiated control strategy with hygienic or other management measures before the farm visit. This effect of reverse causation [33] is well known in cross-sectional observational studies, where the status of a disease and the risk factors are investigated at the same point in time. However, we can only assume that reverse causation is the reason for these confusing results. Hence, further research should be conducted using personal interviews to elicit the true reasons for these results.

#### 4.3.2. Calf Feeding

In the study we presented here, “calf feeding” was the only component of BFM with a statistically significant impact on the MAP status, indicating its importance for future biosecurity and farm management consultancy. A closer investigation of the respective variables indicated that the neglected cleaning of the drinking buckets for different age groups could be a possible source of infection with MAP.

### 4.4. Limits and Strengths

Because only fecal environmental samples were taken, the MAP status might have been estimated as a false negative in some farms, resulting in a biased odds ratio. Therefore, taking individual samples could have led to more precise results for the MAP status. Univariable and multivariable analyses revealed interesting results. However, they could not always identify risk factors with statistical significance due to the comparably small sample size for logistic regression. As these risk factor analyses were only enabled through cooperation with another large project on dairy cattle health in Germany [23] and the study participation was voluntary, we had to cope with the achieved sample size of 457 farms in total which had to be analyzed per region due to different agricultural structures [22]. Only 40% (*n* = 104 out of 260 PraeRi participants) of the addressed farms in the study region south participated, indicating difficulties in recruiting participants for this part of the study. Therefore, a selection bias must be considered here. This could have influenced the results for the study region south. However, the majority of the addressed northern and eastern farms participated in this study, providing more meaningful results. Moreover, all farms were randomly selected, which is a strength of the study.

## 5. Conclusions

We identified several risk factors as known from previous studies from all around the world having a high impact for MAP in German dairy herds. The risk factors did not seem to contribute evenly to MAP infection in all farms, indicating a multifactorial background. However, calf feeding seems to have universal validity in MAP control. Furthermore, the purchase of cattle with an unknown MAP status, the purchase of calves for replacement and hygiene and usage habits of the drinking buckets seemed to have a considerable impact and should be taken into account in veterinary consultancy in Germany as well as in other regions of the world that are structurally comparable with Germany. Nevertheless, a tailored control approach is indicated for individual farms and regions.

## Figures and Tables

**Figure 1 animals-13-01889-f001:**
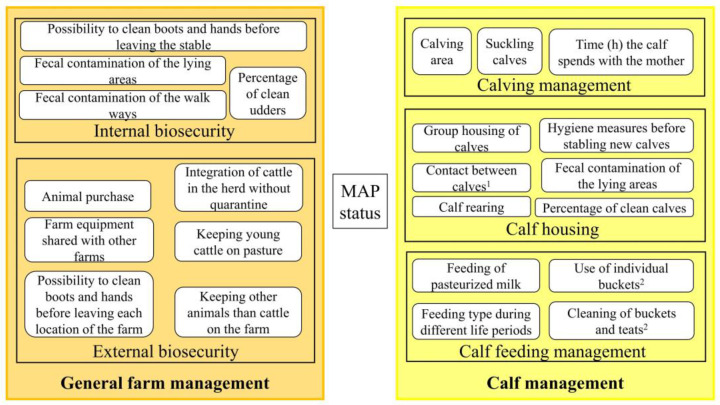
Components of biosecurity and farm management (BFM) for paratuberculosis on a farm with the assigned variables of interest; ^1^ = contact between calves up to two weeks if housing in individual calf igloo; ^2^ = in the case of bucket drinking/automatic feeding.

**Figure 2 animals-13-01889-f002:**
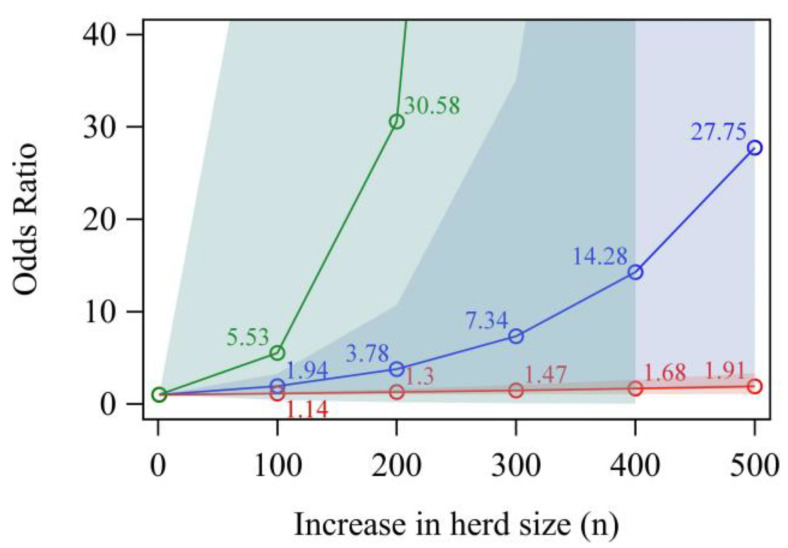
Odds ratio from univariable analysis for quantitative herd size on the MAP status of a farm. Study region north = blue line; study region east = red line; study region south = green line; colored shaded areas = approximate confidence intervals (95% C.I.’s); MAP = *Mycobacterium avium* spp. *paratuberculosis*; *n* = number of cows.

**Table 1 animals-13-01889-t001:** Descriptive and single-factorial analyses of quantitative variables considered to be risk factors for MAP-positive dairy herds (*Mycobacterium avium* ssp. *paratuberculosis* was isolated from at least one environmental sample).

Variable	MAP-Positive Herds				MAP-Negative Herds				OR	LCL-UCL	*p*
	*n*	Mean	Median	CV	*n*	Mean	Median	CV			
Herd characteristics											
Herd size	N: 22 E: 69 S: 3	N:133.64 E:371.14 S: 66	N: 112 E: 250 S: 72	N: 56.84 E: 90.56 S: 37.21	N: 161 E: 101 S: 101	N: 90.40 E:241.74 S: 42.65	N: 73 E: 144 S: 37	N: 72.49 E:136.22 S: 64.28	N: 1.01 E: 1.00 S: 1.02	N: 1.00–1.01 E: 1.00–1.00 S: 0.99–1.04	N: 0.01 E: 0.02 S: 0.18
General farm management											
Internal biosecurity: Percentage of clean udders ^1^	N: 22 E: 69 S: 3	N: 33.82 E: 49.74 S: 62.33	N: 27 E: 51 S: 63	N: 60.32 E: 37.76 S: 22.48	N: 161 E: 101 S: 99	N: 44.42 E: 55.53 S: 38.90	N: 45 E: 55 S: 38	N: 47 E: 35.63 S: 53.48	N: 0.98 E: 0.99 S: 1.05	N: 0.95–1.0 E: 0.97–1.0 S: 1.0–1.1	N: 0.03 E: 0.06 S: 0.08
Calf management											
Calving: Time (h) calf spends with the mother	N: 12 E: 26 S: 1	N: 23.83 E: 14.12 S: 6	N: 12 E: 8.5 S: 6	N:138.27 E:174.74 S: n.p.	N: 90 E: 57 S: 31	N: 55.96 E: 23.53 S: 11.74	N: 12 E: 12 S: 5	N:610.14 E: 150.2 S: 117.27	N: 1.0 E: 0.98 S: 0.95	N: 0.99–1.00 E: 0.96–1.01 S: 0.75–1.21	N: 0.78 E: 0.25 S: 0.69
Calf housing: Percentage of clean calves ^2^	N: 20 E: 69 S: 3	N: 54.8 E: 49.78 S: 62	N: 50.5 E: 50 S: 71	N: 51.52 E: 49.08 S: 41.47	N: 153 E: 97 S: 97	N: 65.7 E: 62.21 S: 65.63	N: 67 E: 67 S: 67	N: 39.34 E: 44.66 S: 45.22	N: 0.99 E: 0.98 S: 1.0	N: 0.97–1.00 E: 0.97–0.99 S: 0.96–1.04	N: 0.09 E: <0.01 S: 0.84

N = Study region north, E = Study region east, S = Study region south, CV = coefficient of variation, OR = Odds Ratio, LCL = Lower Confidence Limit, UCL = Upper Confidence Limit, n.p. = calculation not possible, ^1^ clean udders = udders that were not covered with any dirt, ^2^ clean calves = calves with no or minor dirt located on the abdomen, body side and back side.

**Table 2 animals-13-01889-t002:** Multivariable analyses for different biosecurity components with quantitative herd size.

Component of Biosecurity	Category	OR	LCL	UCL	*p*
First multivariable analysis					
Herd size		N. 1.01 E. 1.00	N:1.00 E: 1.00	N: 1.01 E: 1.00	N: 0.01 E: 0.01
General farm management (GFM)	Good ^a^				N: 0.42 ^b^ E: 0.17 ^b^
	Average	N: 3.68 E: <0.001	N: 0.33 E: <0.001	N: 41.65 E: >999.999	N: 0.3 E: 0.97
	Insufficient	N: 1.62 E: 1.99	N: 0.62 E: 0.98	N: 4.19 E: 4.07	N: 0.33 E: 0.06
Calf management (CM)	Good ^a^				N: 0.91 ^b^ E: 0.49 ^b^
	Average	N: 0.85 E: 1.66	N: 0.21 E: 0.58	N: 3.46 E: 4.73	N: 0.83 E: 0.34
	Insufficient	N: 0.8 E: 1.41	N: 0.29 E: 0.7	N: 2.23 E: 2.86	N: 0.67 E: 0.33
Second multivariable analysis					
Herd size		N: 1.01 E: 1.00	N: 1.00 E: 1.00	N. 1.01 E: 1.00	N: 0.02 E: 0.05
GFM: Internal biosecurity	Good ^a^				E: 0.05 ^b^
	Average	E: 0.46	E: 0.1	E: 2.09	E: 0.31
	Insufficient	E: 1.72	E: 0.52	E: 5.64	E: 0.37
GFM: External biosecurity	Good ^a^				N: 0.63 ^b^ E: 0.42 ^b^
	Average	N: 0.85 E: 1.2	N: 0.26 E: 0.54	N: 2.78 E: 2.65	N: 0.78 E: 0.66
	Insufficient	N: 1.74 E: 0.25	N: 0.47 E: 0.03	N: 6.47 E: 2.37	N: 0.41 E: 0.23
CM: Calving	Good ^a^				N: 0.95 ^b^ E: 0.35 ^b^
	Average	N: 0.75 E: 0.85	N: 0.12 E: 0.21	N: 4.86 E: 3.44	N: 0.76 E: 0.81
	Insufficient	N: 0.78 E: 0.52	N: 0.13 E: 0.13	N: 4.79 E: 2.15	N: 0.78 E: 0.36
CM: Calf feeding	Good ^a^				N: 0.03 ^b^
	Average	N: 5.22	N: 1.25	N: 21.83	N: 0.02
	Insufficient	N: 1.6	N: 0.4	N: 6.34	N: 0.5

^a^ = Reference category, N = Study region north, E = Study region east, OR = Odds Ratio, LCL = Lower Confidence Limit, UCL = Upper Confidence Limit, ^b^ = global *p*-value.

**Table 3 animals-13-01889-t003:** Descriptive and univariable analyses of different biosecurity components considered to be a risk for *Mycobacterium avium* ssp. *paratuberculosis* (MAP)-positive dairy herds (*Mycobacterium avium* ssp. *paratuberculosis* was isolated from at least one environmental sample).

Component of Biosecurity (*n* = Herds with Information Available)	Category	MAP-Positive Herds *n* (%)	MAP-Negative Herds *n* (%)	OR	LCL-UCL	*p*	AFe	AFp
General farm management (GFM) (N: *n* = 183, E: *n* = 170, S: *n* = 104)	Good ^a^	N: 10 (45.45) E: 46 (66.67) S: 3 (100)	N: 89 (55.28) E: 78 (77.23) S: 79 (78.22)			N: 0.56 ^b^ E: 0.27 ^b^		
	Average	N: 1 (4.55) E: 0 (0.0) S: 0 (0.0)	N: 3 (1.86) E: 1 (0.99) S: 1 (0.99)	N: 2.97 E: <0.001	N: 0.28–31.29 E: <0.001–> 999.999	N: 0.44 E: 0.97	N: 0.60 E: n.p.	N: 0.05 E: −0.01
	Insufficient	N: 11 (50.00) E: 23 (33.33) S: 0 (0.0)	N: 69 (42.86) E: 22 (21.78) S: 21 (20.79)	N: 1.42 E: 1.77	N: 0.57–3.53 E: 0.89–3.53	N: 0.78 E: 0.97	N: 0.27 E: 0.27	N: 0.14 E: 0.14
GFM: Internal biosecurity (N: *n* = 183, E: *n* = 170, S: *n* = 104)	Good ^a^	N: 2 (9.09) E: 5 (7.25) S: 1 (33.33)	N: 14 (8.70) E: 10 (9.90) S: 21 (20.79)			N: 0.43 ^b^ E: 0.11 ^b^		
	Average	N: 1 (4.55) E: 6 (8.70) S: 1 (33.33)	N: 25 (15.53) E: 20 (19.80) S: 11 (10.89)	N: 0.28 E: 0.6	N: 0.02–3.37 E: 0.15–2.46	N: 0.23 E: 0.17	N: −2.25 E: −0.44	N: −0.75 E: −0.24
	Insufficient	N: 19 (86.36) E: 58 (84.06) S: 1 (33.33)	N:122 (75.78) E: 71 (70.30) S: 69 (68.32)	N: 1.09 E: 1.63	N: 0.23–5.18 E: 0.53–5.05	N: 0.29 E: 0.06	N: 0.07 E: 0.57	N:0.07 E: 0.52
GFM: External biosecurity (N: *n* = 183, E: *n* = 170, S: *n* = 104)	Good ^a^	N: 13 (59.09) E: 52 (75.36) S: 3 (100)	N:100 (62.11) E: 73 (72.28) S: 73 (72.28)			N: 0.75 ^b^ E: 0.41 ^b^		
	Average	N: 5 (22.73) E: 16 (23.19) S: 0 (0.0)	N: 41 (25.47) E: 22 (21.78) S: 9 (8.91)	N: 0.94 E: 1.02	N: 0.31–2.8 E: 0.49–2.13	N: 0.62 E: 0.24	N: −0.06 E: 0.01	N: −0.02 E: 0.00
	Insufficient	N: 4 (18.18) E:1 (1.45) S: 0 (0.0)	N: 20 (12.42) E: 6 (5.94) S: 19 (18.81)	N: 1.54 E: 0.23	N: 0.46–5.21 E: 0.03–2.00	N: 0.45 E: 0.18	N: 0.31 E: −1.91	N: 0.07 E: −0.04
Calf management (CM) (N: *n* = 183, E: *n* = 170, S: *n* = 104)	Good ^a^	N: 11 (50.00) E: 32 (46.38) S: 3 (100.00)	N: 73 (45.34) E: 52 (51.49) S: 51 (50.50)			N: 0.92 ^b^ E: 0.74 ^b^		
	Average	N: 3 (13.64) E: 9 (13.04) S: 0 (0.0)	N: 23 (14.29) E: 10 (9.90) S: 11 (10.89)	N: 0.87 E: 1.46	N: 0.22–3.37 E: 0.54–3.99	N: 0.95 E: 0.54	N: −0.13 E: 0.20	N: −0.03 E: 0.04
	Insufficient	N: 8 (36.36) E: 28 (40.58) S: 0 (0.0)	N: 65 (40.37) E: 39 (38.61) S: 39 (38.61)	N: 0.82 E: 1.17	N: 0.31–2.16 E: 0.61–2.25	N: 0.8 E: 0.92	N: −0.19 E: 0.09	N: −0.08 E: 0.04
CM: Calving (N: *n* = 183, E: *n* = 170, S: *n* = 104)	Good ^a^	N: 2 (9.09) E: 6 (8.70) S: 0 (0.0)	N: 10 (6.21) E: 5 (4.95) S: 8 (7.92)			N: 0.86 ^b^ E: 0.15 ^b^		
	Average	N: 9 (40.91) E: 33 (47.83) S: 3 (100)	N: 64 (39.75) E: 37 (36.63) S: 63 (62.38)	N: 0.7 E: 0.74	N: 0.13–3.74 E: 0.21–2.66	N: 0.82 E: 0.74	N: −0.35 E: −0.16	N: −0.29 E: −0.13
	Insufficient	N: 11 (50.00) E: 30 (43.48) S: 0 (0.0)	N: 87 (54.04) E: 59 (58.42) S: 30 (29.70)	N: 0.63 E: 0.42	N: 0.12–3.27 E: 0.12–1.5	N: 0.6 E: 0.07	N: −2.34 E: 0.15	N: −1.15 E: 0.14
CM: Calf housing (N: *n* = 183, E: *n* = 170, S: *n* = 104)	Good ^a^	N: 16 (72.73) E: 40 (57.97) S: 2 (66.67)	N:101 (62.73) E: 71 (70.30) S: 73 (72.28)			N: 0.66 ^b^ E: 0.23 ^b^		
	Average	N: 4 (18.18) E: 22 (31.88) S: 1 (33.33)	N: 39 (24.22) E: 21 (20.79) S: 25 (24.75)	N: 0.65 E: 1.86	N: 0.2–2.06 E: 0.91–3.79	N: 0.78 E: 0.26	N: −0.47 E: 0.30	N: −0.09 E: 0.10
	Insufficient	N: 2 (9.09) E: 7 (10.14) S: 0 (0.0)	N: 21 (13.04) E: 9 (8.91) S: 3 (2.97)	N: 0.6 E: 1.38	N: 0.13–2.81 E: 0.48–3.99	N: 0.72 E: 0.98	N: −0.57 E: 0.18	N: −0.06 E: 0.03
CM: Calf feeding (N: *n* = 183, E: *n* = 170, S: *n* = 104)	Good ^a^	N: 3 (13.64) E: 10 (14.49) S: 0 (0.0)	N: 48 (29.81) E: 24 (23.76) S: 26 (25.74)			N: 0.01 ^b^ E: 0.34 ^b^		
	Average	N: 9 (40.91) E: 22 (31.88) S: 1 (33.33)	N: 23 (14.29) E: 28 (27.72) S: 8 (7.92)	N: 6.26 E: 1.89	N: 1.55–25.34 E: 0.75–4.76	N: <0.01 E: 0.35	N: 0.79 E: 0.33	N: 0.59 E: 0.23
	Insufficient	N: 10 (45.45) E: 37 (53.62) S: 2 (66.67)	N: 90 (55.90) E: 49 (48.51) S: 67 (66.34)	N: 1.78 E: 1.81	N: 0.47–6.77 E: 0.77–4.25	N: 0.48 E: 0.39	N: 0.41 E: 0.32	N: 0.32 E: 0.25

^a^ = Reference category, *n* = Herds with information available, N = Study region north, E = Study region east, S = Study region south, OR = Odds Ratio, LCL = Lower Confidence Limit, UCL = Upper Confidence Limit, Afe = Attributable fraction exposed, Afp = Population attributable fraction, ^b^ = global *p*-value.

## Data Availability

The data were made available through internal cooperation. Therefore, any data transfer to interested persons is not allowed without an additional formal contract. Data are available to qualified researchers who sign a contract with the University of Veterinary Medicine Hannover. This contract will include guarantees to the obligation to maintain data confidentiality in accordance with the provisions of the German data protection law. A data access committee will be established on demand. This committee will consist of the authors as well as of members of the University of Veterinary Medicine Hannover. Interested cooperative partners, who are able to sign a contract as described above, may contact Amely Campe, Department of Biometry, Epidemiology and Information Processing University of Veterinary Medicine, Hannover Bünteweg 2, 30559 Hannover, Email: amely.campe@tiho-hannover.de.

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
