# Peer review of "High-Impact Risk Factors for Mycobacterium avium ssp. paratuberculosis in Dairy Herds in Germany"

_animals, 2023, doi:10.3390/ani13121889_

Round 1
Reviewer 1 Report
Dear Authors!
Thank your for this sound work and manuscript!
Some minor recommendations / hints:
Line 19: „These“ not „This“.
Line 33: Comma before „respectively“
Line 74: Better: „…the highest impact on…
Line 207: “33%-quantile” not “quartile” (implies 25, 50 or 75 %)
Line 203: CI 0.43-3.81 but p=0.01?????
Line 219. Why “also”? The comparison is here different than above: Calves from other farms / only own calves versus No calf rearing / only own calves – I am struggling with the logic of “also”!
Lines 226 and 228: For the sake of consistency report CI
Line 232: Maybe better “risk” than “chance”
Lines 236-237: CI 1.09-19.17 but p=0.22?????
Line 389: Not necessarily “underestimated”, depends on distribution of false-negatives, maybe better: “biased”
Line 391: For consistency, “univariable”, not “univariate”
Line 404: From all around the world?Line 19: „These“ not „This“.
Good quality, only minor issues...
Reviewer 2 Report
The manuscript titled “High impact risk factors for Mycobacterium avium ssp. paratuberculosis in dairy herds in Germany” by Krieger et al. provides an in-depth risk analysis for MAP infection to complement a cross-sectional study on paratuberculosis infection in dairy cattle in Germany already published in another manuscript.
This study is well designed and analysed the main recognized risk factors for MAP infection in cattle herds related to German breeding systems. Nonetheless, the study can be overlaid in other countries with similar breeding structure.
Even if the risk factors analysed are known in the transmission and maintenance of MAP, the results obtained confirm that the disease is multifactorial and each risk element must be classified according to priority.
In my opinion, I feel to recommend the publication of this study on “Animals” journal.
To follow, a couple of suggestions to make reading more enjoyable.
Lane 83: I suggest to explain the term “PraeRi”
Lane 102 – 128: I found it difficult to follow this part of the text, I suggest changing the form and schematizing the variables in a more effective and immediate way.
